# Dichotomous Effects of Glypican-4 on Cancer Progression and Its Crosstalk with Oncogenes

**DOI:** 10.3390/ijms25073945

**Published:** 2024-04-02

**Authors:** Victor Chérouvrier Hansson, Fang Cheng, Grigorios Georgolopoulos, Katrin Mani

**Affiliations:** 1Department of Experimental Medical Science, Glycobiology Group, Lund University, Biomedical Center A13, SE-221 84 Lund, Sweden; vi8354ch-s@student.lu.se (V.C.H.); fang.cheng@med.lu.se (F.C.); 2Genevia Technologies Oy, 33100 Tampere, Finland; grigorios.georgolopoulos@geneviatechnologies.com

**Keywords:** glypican 4, TCGA, non-small cell lung carcinoma, glioblastoma, CRISPR/Cas9

## Abstract

Glypicans are linked to various aspects of neoplastic behavior, and their therapeutic value has been proposed in different cancers. Here, we have systematically assessed the impact of GPC4 on cancer progression through functional genomics and transcriptomic analyses across a broad range of cancers. Survival analysis using TCGA cancer patient data reveals divergent effects of *GPC4* expression across various cancer types, revealing elevated *GPC4* expression levels to be associated with both poor and favorable prognoses in a cancer-dependent manner. Detailed investigation of the role of GPC4 in glioblastoma and non-small cell lung adenocarcinoma by genetic perturbation studies displays opposing effects on these cancers, where the knockout of *GPC4* with CRISPR/Cas9 attenuated proliferation of glioblastoma and augmented proliferation of lung adenocarcinoma cells and the overexpression of *GPC4* exhibited a significant and opposite effect. Further, the overexpression of *GPC4* in *GPC4*-knocked-down glioblastoma cells restored the proliferation, indicating its mitogenic effect in this cancer type. Additionally, a survival analysis of TCGA patient data substantiated these findings, revealing an association between elevated levels of GPC4 and a poor prognosis in glioblastoma, while indicating a favorable outcome in lung carcinoma patients. Finally, through transcriptomic analysis, we attempted to assign mechanisms of action to GPC4, as we find it implicated in cell cycle control and survival core pathways. The analysis revealed upregulation of oncogenes, including FGF5, TGF-β superfamily members, and ITGA-5 in glioblastoma, which were downregulated in lung adenocarcinoma patients. Our findings illuminate the pleiotropic effect of GPC4 in cancer, underscoring its potential as a putative prognostic biomarker and indicating its therapeutic implications in a cancer type dependent manner.

## 1. Introduction

Comprehending molecular alterations in cancer is crucial for identifying novel biomarkers and diagnosis tools and to find new therapeutic targets. Tumor microenvironments, including dividing cancer cells, stroma cells, infiltrating inflammatory cells, tumor vascular networks, and a myriad of signaling molecules and extra cellular matrix components secreted by both the tumor and stroma cells, play pivotal roles in cancer development. Among various factors contributing to this complex network, increasing evidence highlights the critical role of the glypican (GPC) family of cell surface heparan sulfate proteoglycans (HSPG) as multifunctional integrators playing an essential role in cellular communication. GPCs engage with a multitude of growth factors, cytokines, and extracellular matrix constituents modulating diverse signaling pathways associated with tumor proliferation, angiogenesis, and metastasis [1,2,3,4]. An accumulating number of in vitro, in vivo, and clinical investigations point out the potential of GPCs in cancer diagnosis and therapy [5,6]. Recent gene expression studies on clinical patient data reveal that members of the GPC family glypican 1–6 (GPC1–6) undergo specific alterations in cancer. In primary solid tumors, *GPC1* and *GPC2* demonstrate significantly higher expression patterns, while *GPC3*, *GPC5*, and *GPC6* exhibit generally lower expression levels as compared to normal healthy tissues [7]. Further, clinical research and patient survival analyses reveal that high GPC1 levels are associated with poor prognosis in a number of cancers including glioblastoma [7,8], pancreatic adenocarcinoma [9], bladder urothelial carcinoma, and liver hepatocellular carcinoma [7]. The inhibition of pathways involved in TGF-β and p38 MAPK signaling has been proposed to be one mechanism of action for GPC1 [7]. Together, evidence supporting the potential role of GPC1 as a novel molecular target in cancer holds promise for GPC1 targeted radioimmunotherapy [10]. Pan-cancer studies on GPC2 have determined its early diagnostic value in 16 kinds of tumors where GPC2 exhibits positive or negative associations with the cancer prognosis [11]. GPC3 is the most extensively investigated member of the GPC family in cancer biology, with a tremendous amount of preclinical and clinical data emphasizing its value in cancer diagnostics and treatment [for review see [6,12]]. A number of reports point out a tumor suppressor function for GPC5 in several cancers, such as non-small cell lung cancer [13], lung adenocarcinoma [14], prostate cancer [15], pancreatic cancer [16], and glioma [17]. In terms of mechanism of action, GPC5-mediated inhibition of pathways involved in epithelial-mesenchymal transition (EMT) has been suggested as a mechanism of action for GPC5 [14,15]. Opposing results show a negative association between *GPC5* expression and the progression of non-small cell lung cancer [18] and breast cancer [19], indicating the necessity of further investigations. An overexpression of *GPC6* has been shown to be correlated with increased patient survival in early-stage ovarian cancer [20]. Also, a biomarker potential of GPC6 in cutaneous melanoma has been reported [21].

Although our understanding of the involvement of GPC4 in cancer is limited, an increasing body of evidence illustrates its essential role in cancer progression. Recent preclinical and clinical reports demonstrate the implications of GPC4 in several cancers, including pancreatic [22], breast [23,24], and colorectal cancer [25]. Further, comprehensive bioinformatic analyses and functional in vitro experiments display a connection between downregulation of GPC4 and the sensitization of pancreatic cancer cells to chemotherapy [22]. The study further reveals that the suppression of GPC4 results in the attenuation of stem cell–like properties of pancreatic cancer via suppression of the Wnt/β-catenin pathway [22]. A grand investigation including clinical breast cancer patients, in vivo, ex vivo, and in vitro studies reveals that GPC4 undergoes downregulation in metastatic tumors and that overexpression of GPC4 induces decreased tumorigenicity, i.e., migration and proliferation, in vitro in metastatic cells as well as in vivo in nude mice [23]. Also, two comprehensive clinical investigations demonstrate increased levels of plasma GPC4 in colorectal and metastatic breast cancer patients [24,25]. The increased plasma level of GPC4 in these cancers was found to be associated with poor patient survival [24,25].

The mechanism of action of GPC4 in cancer is also relatively unknown. Studies show that GPC4 influences various mitogenic signaling pathways including TGF-β [26], TLR4/NF-kB [27], Wnt [22,28,29,30], Mmp14 [31], and FGF2 [32], all of which have the potential to affect cancer progression. GPC4 is also a part of signaling network in CNS, regulating forebrain development by the positive modulation of FGF signaling [33] and promoting fiber sprouting of neurons via mTOR [34]. Further, GPC4 interacts with several components of the synaptic organizing protein complexes, including leucine-rich repeat transmembrane neural proteins (LRRTMs), receptor protein tyrosine phosphatases (RPTPs), and G-protein-coupled receptor 158 (GPR158) [35,36].

In this study, we perform functional genomics assays in glioblastoma and non-small cell lung cancer in vitro models to understand how changes in GPC4 expression affect cancer proliferation, revealing divergent cancer type-dependent effects on proliferation. We then couple these results with a survival analysis of public cancer data, allowing us to link clinical outcomes with *GPC4* expression-associated changes in cancer growth. Finally, a systematic transcriptomic analysis of cancer patients uncovered divergent gene expression profiles between glioblastoma and lung adenocarcinoma subjects, exhibiting pleiotropic effects on the regulation of pathways linked to oncogenic signaling.

## 2. Results

### 2.1. *Wide Range of GPC4 Expression Differences between Normal and Cancer Tissues across TCGA Cancer Types*

A TCGA cancer patient database was used to investigate the direction, magnitude, and significance of GPC4 gene (*GPC4*) expression differences between cancer and normal tissues across 24 TCGA cancer types. The analysis, presented as a volcano plot, exhibits significant differences in *GPC4* expression levels between normal and cancer tissues in a wide range of cancers (Figure 1). Specifically, GPC4 was found to be significantly upregulated in liver cholangiocarcinoma (CHOL), breast invasive carcinoma (BRCA), thymoma (THYM), rectum adenocarcinoma (READ), kidney renal papillary cell carcinoma (KIRP), colon adenocarcinoma (COAD), and glioblastoma (GBM) in comparison to normal healthy tissues (Figure 1, positive change in *GPC4* expression above the *p* value threshold of 0.05). Moreover, *GPC4* expression was found to be significantly downregulated in pheochromocytoma and paraganglioma (PCPG), kidney chromophobe (KICH), kidney renal clear cell carcinoma (KIRIC), cervical and endocervical carcinoma (CESC), bladder carcinoma (BLCA), lung squamous cell carcinoma (LUSC), and head and neck squamous cell carcinoma (HNSC) in comparison to healthy tissues (Figure 1, negative change in GPC4 expression above the *p* value threshold of 0.05). The downregulation of *GPC4* in PCPG and upregulation of *GPC4* in CHOL were shown to be approximately log2-fold of −2 and 2, respectively (4-fold change), while in the other above-mentioned cancer types, statistically significant changes ranged between log2-fold of 0.2 and 2 (Figure 1). Together, these findings emphasize the multifaceted impact of *GPC4* in cancer and stress its potential as a biomarker in particular cancer types.

### 2.2. Cancer Survival Is Associated with GPC4 Expression Levels in a Cancer Type Specific Manner

In order to explore the impact of *GPC4* on cancer survival, we performed a Cox Proportional Hazard (CoxPH) regression against *GPC4* expression levels for each TCGA project. The results identified a significant negative association between *GPC4* gene expression levels and survival in cancer patients with uveal melanoma (TCGA-UVM), pancreatic carcinomas (TCGA-PAAD), lower grade glioma (TCGA-LGG) and brain glioblastoma (TCGA-GBM) (Figure 2A, bold; list of TCGA cancer abbreviations has been provided in Appendix A). Opposite results, with a positive association between *GPC4* gene expression levels and patient survival, were observed in lung adenocarcinoma (TCGA-LUAD) and two different kidney cancers, kidney renal clear cell carcinoma (TCGA-KIRC) and kidney renal papillary cell carcinoma (TCGA-KIRP) (Figure 2A, bold). Further analysis of the association between *GPC4* expression and patients’ survival was accomplished by performing a univariate Kaplan-Meier (KM) survival analysis. Subjects from the aforementioned seven TCGA projects, where significant results were observed in the CoxPH univariate survival analysis for *GPC4* expression levels, were stratified into 3 bins (low, medium, and high) based on their *GPC4* expression levels (please see Methods). As presented in Figure 2B, Kaplan-Meier curves reveal a statistically significant correlation between higher levels of *GPC4* expression and a poor prognosis in uveal melanoma, pancreatic carcinoma, glioma, and glioblastoma cancers. Additionally, we detected a statistically significant association between higher levels of GPC4 and favorable outcomes in lung carcinomas and kidney cancers (Figure 2B). These observations highlight the significance of GPC4 as a survival prognostic marker.

### 2.3. Suppression of GPC4 Attenuates Proliferation of Glioblastoma and Augments Proliferation of Lung Adenocarcinoma Cells

Our survival analysis of *GPC4* expression data in TCGA displayed an association between high levels of *GPC4* expression and poor prognoses in glioblastoma patients. In contrast, in lung adenocarcinoma patients, low levels of GPC4 were associated with poor prognoses. In order to explore the impact of GPC4 on the progression of these cancer types, the expression of *GPC4* was transiently disrupted by a *GPC4*-targeted CRISPR double nickase consisting of *GPC4*-specific 20 nt guide RNA sequences derived from the GeCKO (v2) library (CRISPR/Cas9 GPC4), and the effects on proliferation of different cancer cell lines isolated from cancer patients was investigated. Two glioblastoma cell variants isolated from a 75 year old female (SNB-75 cells) and a 67 year old female (SF-295 cells) and a non-small cell lung adenocarcinoma cell variant isolated from a 62 year old male (HOP-92) were therefore either treated with CRISPR/Cas9 GPC4 to disrupt GPC4 gene expression or a control double nickase plasmid with gRNA not targeting any gene (CRISPR control). The suppression of *GPC4* expression was determined by immunofluorescence microscopy, which indicated a significant decrease of GPC4 levels upon CRISPR/Cas9 GPC4 transfection (Figure 3A and insets). Further, slot blot assays confirmed a considerable decrease of immunoreactivity with the GPC4 antibody in CRISPR/Cas9 GPC4 transfected cells compared to CRISPR controls, though the GPC4 signal was not completely obliterated (Appendix A).

Next, the effect of CRISPR/Cas9 GPC4 on the proliferation of SNB-75 and SF-295 glioblastoma cells and HOP-92 non-small cell lung adenocarcinoma cells was investigated. Control groups for each cancer type were treated with the CRISPR control. A series of untreated cells grown in only culture medium were included to monitor cell proliferation in the absence of treatments. After 3 days of culturing, the cells were fixated in glutaraldehyde and stained with crystal violet, followed by lysis in Triton X-100 (see Methods). By measuring bound crystal violet dye at A595 nm using a Byonoy micro plate reader, mean staining intensities were obtained and the background staining of empty plastic plates was subtracted. The results displayed that targeting GPC4 with CRISPR/Cas9 GPC4 attenuates the proliferation of glioblastoma cells and augments the proliferation of non-small cell lung adenocarcinoma cells in comparison with untreated cells and CRISPR controls (Figure 3B). With the mean staining intensity of the CRISPR control cells set at 100%, CRISPR/Cas9 GPC4 cells had the following mean relative cell counts; SNB-75: 34.0% (SE 6.2%), SF-295: 33.3% (SE 6.7%), and HOP-92: 170.2% (SE 14.4%) (Figure 3B). We further determined *p* values in duplicate experiments using a two-tailed Student’s *t*-test with equal variance, *n* = 6 for each experiment. The decrease in relative cell count was shown to be significant for both glioblastoma cancers (*p* values; SNB-75: 0.01 and SF-295: 0.00007). Further, the increase in relative cell count was shown to be significant in the HOP-92 lung adenocarcinoma cells (*p* values; HOP-92: 0.0001). Also, with the untreated mean cell count set at 100%, the mean relative cell count for CRISPR/Cas9 GPC4 cells displayed a significant decrease in proliferation rate for SNB-75: 48.1% (SE 7.6%) and SF-295: 36.3% (SE 4.5%) cells (*p* values; SNB-75: 0.00008 and SF-295: 0.00000002) and a significant increase of the proliferation rate of HOP-92: 162.0% (SE 12.6%) (*p* value; HOP-92: 0.0005).

### 2.4. Overexpression of GPC4 Augments Proliferation of Glioblastoma and Attenuates Proliferation of Lung Adenocarcinoma Cells

In order to elucidate the effect of overexpression of *GPC4* on the proliferation of glioblastoma and non-small cell lung carcinoma cells, SNB-75, SF-295 and HOP-92 cells were either transfected with pCMV3-C-GFPSpark GPC4 overexpression vector (overexpression GPC4) or pCMV3-C-GFPSpark negative control vector (overexpression control) or were left untreated to monitor their proliferation rate in the absence of any treatment. Immunofluorescence microscopy and immunoblot slot blot assays were used to monitor *GPC4* overexpression. Measurements of the GPC4 signal intensity in 4 representative immunofluorescence images displayed significant increases of GPC4 levels (Figure 3C and insets), and immune slot blot assays confirmed considerable enhancements of immunoreactivity with GPC4 antibody in cells transfected with overexpression vector compared to control vector (Appendix A).

Proliferation studies exhibited that increasing the expression of *GPC4* results in increases in the proliferation rates of SF-295 and SNB-75 cells and a decrease in the proliferation rate of HOP-92 (Figure 3D). With the mean staining intensity of overexpression control cells set at 100%, overexpression *GPC4* cells had the following mean relative cell counts; SNB-75: 190.1% (SE 2.84%), SF-295: 117.4% (SE 2.6%) and HOP-92: 33.5% (SE 10.7%). Although increasing the expression of GPC4 resulted in increased proliferation in SF-295 and SNB-75 cells, only the results for SNB-75 cells proved significant (*p* values; SNB-75: 0.01 and SF-295: 0.48). The decrease in relative cell count was significant in lung adenocarcinoma cells (*p* values; HOP-92: 0.00004). Transfection with the overexpression control plasmid showed no significant effect on the proliferation rate compared to untreated cells (Figure 3D). With the untreated mean cell count set at 100%, the mean relative cell counts for overexpression *GPC4* cells were as follows; SNB-75: 202.2% (SE 30.5%), SF-295: 170.1% (SE 24.3%), and HOP-92: 38.7% (SE 2.9%). When *p* values were determined using a two-tailed Student’s *t*-test with equal variance, *n* = 6 in double experiments, the increase in relative cell count turned out to be significant for both glioblastoma cell lines, and the decrease in relative cell count was significant in the lung adenocarcinoma cell line (*p* values; SNB-75: 0.008, SF-295: 0.015, HOP-92: 3 × 10^−10^).

### 2.5. Overexpression of GPC4 in GPC4-Knockdown Glioblastoma Cells Resulted in Restored Proliferation Rate

In order to confirm that the observed decrease in the proliferation rate in CRISPR/Cas9 GPC4 knocked-down glioblastoma cells was due to a loss of GPC4 function, we conducted a series of rescue experiments, aiming to restore GPC4 function by transfecting *GPC4* knocked-down cells with a *GPC4* overexpression vector. In these experiments, the SNB-75 and SF-295 cells were treated with CRISPR/Cas9 GPC4 knockdown plasmid or a scrambled CRISPR control vector for 3 days and were then fixated in glutaraldehyde to confirm the effects of *GPC4* knockdown. Further, a series of CRISPR/Cas9 GPC4 SNB-75 and SF-295 cells were further transfected with *GPC4* overexpression control vector or *GPC4* overexpression vector for another 3 days followed by fixation in glutaraldehyde. All cells were then stained with crystal violet as described in Materials and Methods, and the level of staining was measured at A595 nm using a Byonoy micro plate reader. Background staining was subtracted. As expected, the knockdown of *GPC4* resulted in a significant decrease of proliferation of SNB-75 and SF-295 cells in comparison with the CRISPR control (Figure 4, *p* Values; SNB-75: 0.000000004 and SF-295: 0.000000001, respectively). In both the SNB-75 and SF-295 cells, CRISPR/Cas9 GPC4 knocked-down cells’ subsequent transfection with *GPC4* overexpression vector resulted in a significant increase of proliferation compared to cells transfected with the overexpression control vector (*p* values; SNB-75: 0.00007, SF 295: 0.000008, two-tailed Student’s *t*-test with unequal variance, *n* = 6). SF-295 cells showed no significant difference in proliferation between CRISPR/Cas9 GPC4 knocked-down cells and cells transfected with overexpression control vector. However, the same comparison showed a slight but significant difference in staining intensity for SNB-75 cells (Figure 4).

Taken together, these results indicate that the attenuation of the proliferation of SNB-75 and SF-295 cells upon CRISPR/Cas9 GPC4 knockdown was due to a loss of GPC4 function, as restoring GPC4 via an overexpression vector restored the proliferation rate to the level of the control.

### 2.6. Upregulation of GPC4 Activates Proto-Oncogenes in Glioblastoma but Not in Lung Adenocarcinoma

In order to gain insights into the potential mechanisms through which GPC4 can confer cancer phenotypes, we systematically compared differentially expressed genes between *GPC4*-low and *GPC4*-high subjects across TCGA cancer types. Ingenuity Pathway Analysis (IPA) on 352 differentially expressed genes (Benjamini-Hochberg adjusted *p* value < 0.05 and absolute log2 Fold Change > 0.58) found in at least 10 cancer types revealed their involvement in mechanisms associated with migration, invasion, and vascularization, key features of cancer progression (Figure 5A). More specifically, pathways involved in cell cycle control, such as S100 family signaling and some immunological aspects of cancer, namely pathogen-induced cytokine storm signaling, IL-12 signaling, and production of macrophages were pointed out (Figure 5B and Table 1). Additionally, we observed enrichment in genes that take part in the signaling pathway of phosphatase and tensin homologue (PTEN), known to suppress tumor growth and metastasis via inhibition of PI3K/Akt/mTOR [37]. Notably, aberrant PTEN signaling has been implicated as one of the key mechanisms in both lung cancer, lung metastasis, and in glioblastoma [37,38]. Other enriched terms include signaling pathways in embryonic and brain development, whose deregulation is often implicated in cancer development [39].

Finally, in order to better understand the contrasting effects of *GPC4* upregulation between lung adenocarcinoma and glioblastoma observed in our in vitro experiments, we systematically studied genes with divergent gene expression profiles between TCGA-LUAD and TCGA-GBM subjects (Appendix A). We identified 13 genes overexpressed (log2 fold change > 0.58 and BH adjusted *p* value < 0.05) in *GPC4*-high TCGA-GBM subjects over the *GPC4*-low but downregulated (log2 fold change < −0.58 and BH adjusted *p* value < 0.05) in the respective TCGA-LUAD cohort (Figure 6A, red). Among these, genes positively associated with mitogenic activities and cancer progression include FGF5, FOSL1, and FST (endcoding follistatin that belongs to the TGF-β superfamily). Additionally, we identified ITGA5 integrin, which is involved in cell adhesion, and the EMT and PTGES genes, encoding prostaglandin E synthase, a known promoter of inflammation and immune response (Figure 6A). Importantly, we identified DKK1 gene encoding Dickkopf-1 which is a secretory antagonist of the Wnt signaling pathway. Finally, a set of recently discovered cancer-associated lncRNAs, namely RPSAP52 (Ribosomal protein SA pseudogene 52), LINC00941, and LINC02577 were identified (Figure 6A) [40].

Following further exploration by subjecting these genes to pathway analysis, we discovered that these genes represent mechanisms involving primarily Wnt, FGFR, and the receptor protein serine/threonine kinase signaling pathways and cancer (Figure 6B). Moreover, we found enrichment of the mechanisms associated with endodermal differentiation, which is directly regulated by Wnt. Finally, we found enrichment for EMT (endothelial to mesenchymal transition), a well described feature of glioblastoma progression [41]. Conclusively, these results provide insights into potential mechanisms that explain the divergent proliferation outcomes of *GPC4* upregulation between glioblastoma and lung adenocarcinoma.

## 3. Discussion

GPCs have emerged as pivotal players in the intricate landscape of cancer biology. Their multifaceted interactions with growth factors and various components of the extracellular matrix position these macromolecules at the crossroads of critical signaling pathways involved in malignant behavior. The influence of GPCs extends beyond the boundaries of individual tumors. They also contribute to the intricate interplay between cancer cells and their microenvironment. Within the GPC family, GPC4 stands out as a less studied but intriguing member, warranting a closer examination of its role in cancer progression. In this investigation, we have systematically exploited the influence of GPC4 in cancer by creating a comprehensive profile of its expression pattern in normal and malignant tissues across a broad range of cancers. Utilizing clinical patient data from TCGA cohorts, we discovered that GPC4 exhibits a cancer-specific gene expression pattern associated with cancer progression and patient survival (Figure 2). A CoxPH analysis fitted against *GPC4* expression values exhibited a statistically significant association between high GPC4 levels and a poor prognosis in glioma, glioblastoma, pancreatic carcinoma, and uveal melanoma, suggesting its potential as a prognostic factor. Furthermore, elevated levels of GPC4 were shown to be associated with favorable outcomes in lung carcinomas and kidney cancers, indicating its multifaceted role in cancer. A KM survival analysis also demonstrated that among patients with lung carcinomas and kidney cancers, high *GPC4* expression was associated with better overall survival, while in patients with glioma, glioblastoma, pancreatic carcinoma, and uveal melanoma, those with high *GPC4* expression had shorter survival times. Also, experimental in vitro investigations knocking out *GPC4* expression with CRISPR/Cas9 in cell lines originating from cancer patients revealed the divergent effects of GPC4 in glioblastoma and non-small cell lung carcinoma, i.e., attenuation of proliferation in glioblastoma and promotion of proliferation in non-small cells lung adenocarcinoma cells. Opposite effects were obtained upon *GPC4* overexpression, inducing proliferation of glioblastoma and suppressing proliferation of non-small cell lung adenocarcinoma cells (Figure 3). Further, overexpression of GPC4 in *GPC4* knocked-down glioblastoma cells restored their proliferation rate, indicating that the attenuation of proliferation in SNB-75 and SF-295 cells upon CRISPR/Cas9 GPC4 knockdown is attributable to the loss of GPC4 function, which suggests a mitogenic effect of GPC4 in this cancer type. Contradictory results on the effects of GPC4 on cancer have been previously reported in breast cancer [42]. In this study, high levels of GPC4 were shown to be associated with longer relapse-free time in unclassified breast cancer, whereas in estrogen receptor negative and HER2-positive breast cancer low GPC4 levels led to a longer relapse-free time [42]. These results indicate the dichotomous nature of GPC4, acting as a tumor promoter or suppressor depending on the cancer type. Further, the distinct and diverse effects of GPC4 on the progression of different cancers propose its pleiotropic effect on the tumor microenvironment and oncogenic signaling.

Structurally, GPC4 consists of a core protein substituted with heparan sulfate chains (HS), all anchored to the cell membrane via a glycosylphosphatidylinositol (GPI) linkage that anchors GPC4 to the cytoplasmic surface of cellular membranes. The functional relevance of GPC4 can be attributed to both its core protein and its HS chains, which affect its interactions with various molecules in terms of selectivity and binding affinities, thereby contributing to the complexity of cellular communication and fine-tuning signaling pathways [43]. The HS display a tremendous structural diversity as a result of a tightly controlled biosynthetic pathway, exhibiting differential regulation in different organs, stages of development, and pathologies, including cancer. As an attempt to gain insights into the potential mechanisms through which GPC4 confers cancer, we performed an IPA pathway analysis of genes that were differentially expressed between *GPC4*-low and *GPC4*-high patients across TCGA cancer types. The analysis revealed the genes’ involvement in mechanisms associated with mitogenic proliferation and immunological combat of cancer, including S100 protein family signaling pathways involved in cell cycle control, IL12 induced immune cell activation of macrophages, and the chemotaxis of NK cells and T-Cells (Figure 5) [44]. Notably, the S100 family of proteins are widely expressed low molecular weight EF-hand calcium-binding proteins involved in numerous cellular processes, such as cell proliferation, differentiation, apoptosis, inflammation, cytokine signaling, and immune response [45]. Also noteworthy is the fact that IL-12 has been shown to be an anti-tumor cytokine that plays a multifaceted role in cancer. It has been shown to display a stronger therapeutic effect than traditional chemotherapy with paclitaxel and cisplatin in lung cancer [46]. The IPA analysis further unveiled enrichment of pathways associated with brain development, synaptogenesis, and axonal guidance, providing potential explanations for the impact of GPC4 in glioma and glioblastoma, which originate from neuronal and non-neuronal stem cells. Markedly, GPC4 is highly expressed and secreted by cortical astrocytes and neural stem cells in the ventricular zone [47]. The CNS neuronal stem cells have the potential to differentiate into neurons, astrocytes, and oligodendrocytes during postnatal development and in adult CNS.

Systematic explorations of genes with divergent expression profiles between glioblastoma and lung adenocarcinoma patients revealed upregulation of a series of genes involved in cell proliferation, differentiation, adhesion, and transformation in glioblastoma; the same genes were downregulated in lung adenocarcinoma (Figure 6). Among these genes, both FST (which belongs to the TGF-β superfamily) and FGF5 are proto-oncogenes, activating serine/threonine kinase and tyrosine kinase receptors, respectively. Previous in vitro studies have shown binding of GPC4 to FGF proteins [32] and the effects of GPC4 in epithelial integrity and MET/EMT by enabling TGF-β sensing [26]. However, a connection between GPC4 and FGF5 or TGF-β in cancer patients has never been reported before. Structural and functional studies have revealed that FGF signal transduction requires the association of FGF with its receptor tyrosine kinase (FGFR) and HSPG in a specific complex on the cell surface [48]. The complex formation results in FGFR dimerization and subsequent signal transduction. Direct involvement of the HS chains in the molecular association between FGF and its receptor has been shown to be essential for biological activity. However, in pancreatic cancer, a core protein-dependent mitogenic response to FGF has been demonstrated [49]. In terms of the mechanisms involved in cell adhesion, EMT transformation, and cancer invasiveness, we discovered that the ITGA5 gene, which encodes integrin alpha 5, is upregulated in glioblastoma but downregulated in lung adenocarcinoma. Interestingly, high ITGA5 expression has been shown to be positively related to aggressive clinicopathological features and poor survival in glioma patients [50]. Also, the PTGES gene, encoding prostaglandin E synthase (which plays a key role in inflammation and immune response) was shown to be dysregulated in a comparison between glioblastoma and lung adenocarcinoma. Further exploration of these data by pathway enrichment analysis revealed mechanisms associated with the Wnt signaling pathway and EMT to be discordantly activated in these cancers. Existing evidence derived solely from experimental studies has demonstrated physical binding of GPC4 to Wnt ligands and its regulatory role in Wnt signaling [22,28,29,30]. Differentially expressed GPC4 can undergo shedding from cell surfaces by the action of either phospholipases or proteases [47]. Shedding of GPC4 may serve as another mechanism to control mitogenic signaling. The released GPC4, which still exhibits biological activity, is detectable in serum, making it a potential diagnostic and prognostic marker. Two clinical studies have already shown increased plasma levels of GPC4 in metastatic colorectal and breast cancer patients to be associated with poor patient survival, proposing its biomarker potential in these cancers [24,25].

Our findings unravel the divergent effects of GPC4 on cancer progression and clinical outcome, particularly in glioblastoma and non-small cell lung cancer, and illuminates the pleiotropic effect of GPC4 on proto-oncogene singling pathways that govern mitogenic behavior. Understanding the intricate effects of GPC4 on neoplastic behavior provides insights into its potential as a prognostic factor and therapeutic target in specific cancer types. Further research is essential to unravel the intricate molecular interactions of GPC4 in different cancer contexts, paving the way for a better understanding of its therapeutic implications and diagnostic value.

## 4. Materials and Methods

### 4.1. TCGA Data Pre-Processing

R package TCGAbiolinks was used to download clinical data and harmonized gene expression data from NCI’s Genomic Data Commons (GDC) TCGA [51,52,53]. The data underwent all further analysis by R, v. 4.0.6 [54]. Expression of the GPC4 gene (HTSeq-Count) was acquired from 33 cancer types (Appendix A). Before conducting survival analyses, the gene counts were logarithmized to base 10 and standardized.

### 4.2. Univariate Survival Analysis

A univariate approach utilizing the R packages survival, v. 3.2-7 [55] and survminer, v. 0.4.8 [56] were used to investigate association between GPC4 gene expression and overall survival as described previously [7]. Data from primary tumors and from patients with available statistics for survival time and vital status were incorporated. The *GPC4* gene was tested against each cancer type individually. To identify relationships between *GPC4* gene expression and survival, a continuous univariate CoxPH proportional hazards model was fitted for GPC4. Next, in all significant relationships between cancer type and the *GPC4* gene, subjects were stratified based on *GPC4* expression levels into “Low” when gene expression level was <25th %, “Medium” (for gene expression levels between ≥25th % and <75th %) and “High” (when gene expression level ≥ 75th %), and a KM univariate model was fitted. Function ggsurvplot was used to plot the KM curves.

### 4.3. Multivariate Cox PH Survival Analysis

A multivariate Cox PH survival analysis was performed, testing survival against the *GPC4* gene across the three gene expression strata, as described before [7]. Cox regression was run using function coxph. The function ggforest was used to generate forest plots illustrating the hazard ratios of each variable.

### 4.4. Differential Expression (DE) Analysis

The DE analysis included subjects from 10 cancer types or subjects with lung adenocarcinoma and glioma cancers (adenocarcinomas: TCGA-COAD, TCGA-LUAD, TCGA-PAAD, TCGA-PRAD, TCGA-READ, TCGA-STAD and gliomas: TCGA-GBM, TCGA-LGG), where *GPC4* expression levels displayed significant results in the CoxPH univariate survival analysis. For each cancer type the HTSeq gene counts were obtained. Based on *GPC4* expression levels, the samples (subjects) were stratified into two groups. Subjects in whom the *GPC4* counts were below the 25th percentile were labeled as “*GPC4*-low” while subjects with *GPC4* counts above the 75th percentile were labeled as “*GPC4*-high”. For each cancer type, DE analysis was performed using the DESeq2 package [57]. After pairwise Wald tests between the *GPC4*-low and *GPC4*-high groups, differentially expressed genes were filtered for Benjamini-Hochberg adjusted *p* value < 0.05 and absolute log2 Fold Change > 0.58 (1.5 fold).

### 4.5. Pathway Analysis

IPA core analysis was applied to differentially expressed genes between groups of *GPC4*-low and *GPC4*-high patients (significance thresholds: adjusted *p* value < 0.05 and |log2 foldchange| > 0.58) in at least 5 TCGA-studied cancer types, comprising a total of 4043 genes and 10 studied TCGA cancer types, comprising a total of 351 genes or genes that were found DE in at least 3 glioma or adenocarcinoma cancers (adenocarcinomas: TCGA-COAD, TCGA-LUAD, TCGA-PAAD, TCGA-PRAD, TCGA-READ, TCGA-STAD and gliomas: TCGA-GBM, TCGA-LGG, 430 genes) [58]. Differentially expressed genes was used for canonical pathway enrichment analysis. A Right-tailed Fisher’s Exact Test was used to calculate the significance values (*p* value of overlap) for the IPA Canonical Pathways. The *p* values were adjusted for multiple testing using the Benjamini-Hochberg correction. A ratio was calculated of the number of DE molecules associated with a given pathway divided by the total number of molecules in the reference set that map to the pathway. IPA also calculated for each pathway a *z*-score that predicted pathway activation (if positive) or inhibition (if negative). The *z*-score is calculated by comparing the dataset fold changes under analysis with the canonical pathway patterns in the IPA Knowledge Base. *z*-scores of ≥2 or ≤−2 are considered significant, and no *z*-score annotation indicates either a zero (or very close to zero) *z*-score or that the given pathway is ineligible for a prediction. Significant canonical pathway terms were filtered for BH adjusted *p* value < 0.05. The IPA Networks algorithm generated the interaction networks of the input DE molecules, scoring the networks based on the count of network eligible molecules that they contained (molecules with known scientific evidence of directly or indirectly interacting with other molecules in the Ingenuity Knowledge Base). The score was based on the hypergeometric distribution and was calculated with the right-tailed Fisher’s Exact Test; the higher the score, the lower the probability of finding the observed number of the input dataset molecules in a given network by random chance. In addition to IPA, genes differentially expressed between TCGA-LUAD and TCGA-GBM were subjected to pathway analysis using the R interface of the enrichr database [59].

### 4.6. Cell Culture

Two glioblastoma cell variants isolated from a 75 year old female (SNB-75 cells) and a 67 year old female (SF-295 cells) were obtained from National Cancer Institute (NCI), Rockville, MD, USA (vial designations #0507820 and #0507441, respectively). Non-small cell lung adenocarcinoma cells isolated from a 62 year old male (HOP-92) was purchased from NCI (vial designation #0507454; NCI, Rockville, MD, USA). Authentication and certificate analyses were provided by NCI and ATCC. The cells were cultured following the manufacturer’s instructions. Additionally, all cancer cells underwent routine treatment with a mycoplasma removal agent for one week following thawing of the frozen cells (cat# 3050044; MP Biomedicals, Eschwege, Germany).

### 4.7. Transfection of Human GPC4 Targeting CRISPR/Cas9

Human GPC4 targeting CRISPR/Cas9 (cat# sc-405200-NIC, Santa Cruz Biotechnology, Dallas, TX, USA), comprising a pair of human *GPC4* targeted CRISPR/Cas9 knockout plasmids and its control plasmid (cat# sc-437281, Santa Cruz Biotechnology, Dallas, TX, USA) not targeting any specific gene (CRISPR control) were purchased from Santa Cruz Biotechnology. These plasmids both express a GFP marker for a visualization of transfection. The cells were transiently transfected with CRISPR/Cas9 GPC4 or the CRISPR control vector for 48–72 h according to the manufacturer’s instructions. Transfections were visualized through detection of GFP by fluorescence microscopy. Knockout of *GPC4* was assessed by analyzing levels of GPC4 in 4–5 samples after immunofluorescence staining with a GPC4 polyclonal antibody (pAb GPC4; Cat# PA5-115301; Invitrogen, Lund, Sweden). Suppression of *GPC4* was confirmed by slot blotting of cell extracts using a GPC4 polyclonal antibody (pAb GPC4; Cat# PA5-115301; Invitrogen, Lund, Sweden). The slot blots underwent stripping and were subsequently reprobed with α-tubulin antibody (cat# A-11126; Molecular Probes, Eugene, OR, USA) to serve as a loading control.

### 4.8. Overexpression of GPC4 Using Ectopic Expression of Green Fluorescent Protein-Tagged Gpc4 (GFP-GPC4)

Transfection was performed using either a human *GPC4* overexpression plasmid containing the C-GFPSpark tag (overexpression *GPC4*) (cat# HG10090-ACG; Sino Biological, Düsseldorf, Germany) or a non-specific control plasmid with the C-GFPSpark tag and not targeting any known gene (overexpression control) (cat#CV026; Sino Biological, Düsseldorf, Germany). The cells were transiently transfected with the *GPC4* overexpression vector or the overexpression control vector for 48–72 h following the protocol for FuGENE 6 transfection provided by Promegas (cat# E2691; Promega Biotech AB, Stockholm, Sweden). Transfections were confirmed through the detection of GFP using fluorescence microscopy. Overexpression of *GPC4* was measured by analyzing levels of GPC4 in 4–5 samples after immunofluorescence staining with a GPC4 polyclonal antibody (pAb GPC4; Cat# PA5-115301; Invitrogen). Overexpression of *GPC4* was confirmed by slot blotting of cell extracts using a GPC4 polyclonal antibody (pAb GPC4; Cat# PA5-115301; Invitrogen, Lund, Sweden). As a loading control, stripped blots were probed with an α-tubulin antibody (cat# A-11126; Molecular probes, Eugene, OR, USA). The stripped blots were also reprobed with a secondary antibody to control background binding.

### 4.9. Fluorescence Imaging

GPC4 expression was evaluated using fluorescence imaging as previously described (7). Specifically, cells transfected with the CRISPR control, CRISPR/Cas9 GPC4, overexpression control, or GFP-GPC4 vectors were washed extensively with phosphate buffer saline (PBS), followed by fixation with acetone in 2 min. The cells were then incubated for about 24 h with rabbit polyclonal anti-human GPC4 antibody (Catalog # PA5-115301; Invitrogen, Lund, Sweden). After thorough PBS washes, the cells were incubated with goat anti-rabbit IgG, Alexa Fluor™ 594 (cat# A-21208; Molecular Probes, Eugene, OR, USA) for 4 h. Nuclei were visualized through DNA staining with 4′,6-diamidino-2-phenylindole (DAPI). The anti-human GPC4 antibody was omitted in the controls. Analysis of the fluorescent images was accomplished using an AxioObserver fluorescence microscope from Carl Zeiss equipped with a 63X/1.25 Oil M27 EC Plan-Neofluar objective and Axiocam 305 mono Camera. All images were scanned using identical exposure settings. Multitrack acquisition and sequential excitation of fluorophores were used to minimize channel crosstalk. The entire slide was scanned during microscopy, and immunofluorescence images were captured at 20× and 60× magnification.

### 4.10. Slot Blot Assay

About 2 × 105 cells cells were lysed in a radio-immunoprecipitation assay buffer (RIPA) containing 0.1% *w*/*v* SDS, 0.5% *v*/*v* Triton X-100, and 0.5% *w*/*v* sodium deoxycholate in PBS, supplemented with a cocktail of proteinase inhibitors (cOmplete mini, cat# 11836153001; Roche, Solna, Sweden). The lysis was performed by shaking for 10 min at 4 °C. Equal volumes of the lysed samples were loaded onto PVDF membranes using a slot blot apparatus. The PVDF membranes were then incubated with an anti-GPC4 antibody with a dilution of 1:500 overnight at 4 °C or for 5 h at room temperature. Subsequently, the membranes were extensively washed with PBS containing 0.5% Tween-20 and treated with horseradish peroxidase-conjugated anti-rabbit IgG with a dilution of 1:500 (cat# 170-6515; Bio-Rad, Hercules, CA, USA). A Pierce fast western blot kit was used to develop the membranes (cat# 35050, Thermo Scientific, Lund, Sweden). A Cytiva ImageQuant 500 chemiluminescence detector was used for visualization of the slot blot bands. The intensity of the bands was quantified using GelAnalyzer, version 19.1. For the loading control, PVDF membranes were stripped in Restore™ PLUS Western Blot Stripping Buffer (cat#46430; Life Technologies, Lund, Sweden). The stripped membranes were then incubated with anti-α-tubulin from Molecular Probe (cat# A-11126, Molecular probes, Eugene, OR, USA), followed by treatment with HRP labeled anti-mouse IgG from Bio-Rad (cat# 172-1011Bio-Rad, Hercules, CA, USA). In the negative controls the GPC4 antibody was excluded.

### 4.11. Proliferation Rate Assay Using Crystal Violet

The proliferation assay methodology has been previously outlined [60]. Dissociation of confluent cells was performed using TrypLETM (cat# 12604-021; Thermo Scientific, Lund, Sweden), and the cells were seeded at a plating density of 50,000 cells/well in 96-well microculture plates for 24 h. The cells were then either left untreated or treated with the CRISPR control, CRISPR/Cas9 GPC4, overexpression control, or GFP-GPC4 vectors for 2–3 days. Controls, including untreated cells and blanks containing only medium, were incorporated. Subsequently, the cells were fixed with 0.25% (*v*/*v*) glutaraldehyde in Hanks’ balanced salt solution for 30 min, and the cell density was further determined by nuclear staining with 0.1% (*v*/*v*) crystal violet (cat# C6158; Sigma-Aldrich, Solna, Sweden) for 30 min. After extensive washing, the cells were lysed in 1% (*v*/*v*) Triton X-100 for 1 h, and the amount of bound dye was measured at A595 nm using a Byonoy microplate reader (cat# ABSMHA01; Absorbance 96 compact reader, Byonoy, Hamburg, Germany). Cell proliferation rates were determined using Byonoy Absorbance 96 software, and relative cell numbers were calculated as a percentage of untreated cells or controls (CRISPR control or overexpression control) using MS Excel. Results are presented in graphs based on experiments performed in duplicate, with *n* = 6 in each experiment and data points expressed as means ± SE.

### 4.12. Statistical Analyses

Each data point in the graphs is expressed as means ± standard error (SE), with a sample size (*n*) of 6 for each experiment. Statistical analyses involved two-group comparisons conducted through unpaired two-tailed Student *t*-tests, considering unequal variances. Statistical significance was defined as error probabilities (*p*) ≤ 0.05. Summary of *p* values: ns (not significant, *p* > 0.05), * *p* ≤ 0.05, ** *p* ≤ 0.01, *** *p* ≤ 0.001, **** *p* ≤ 0.0001.

## Figures and Tables

**Figure 1 ijms-25-03945-f001:**
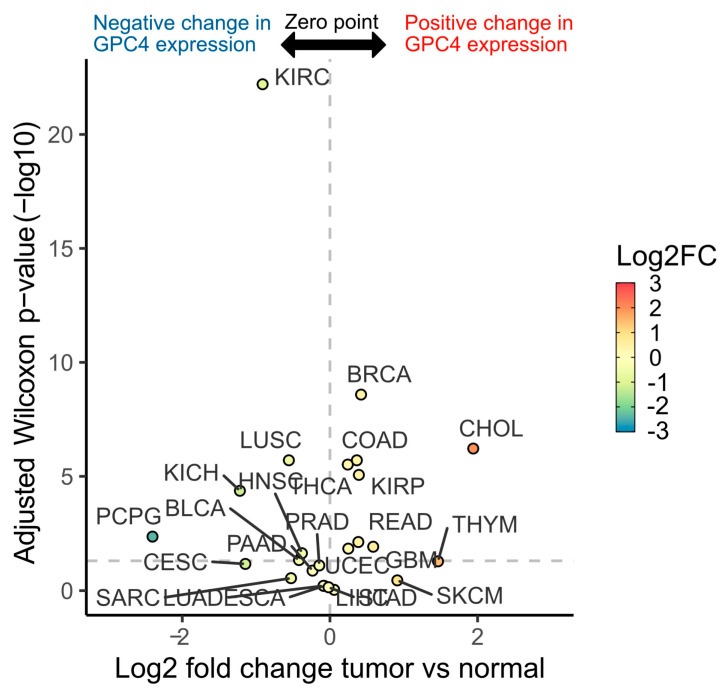
GPC4 displays a wide range of expression differences between normal and tumor tissues across TCGA cancer types. Volcano plot depicts median log2 fold change (*x*-axis and color scale) of GPC4 expression between tumor and normal subjects across 24 TCGA cancer types. Adjusted Wilcoxon rank sum test *p* values (−log10) are shown on the *y*-axis. Horizontal dashed line denotes the 0.05 *p* value threshold.

**Figure 2 ijms-25-03945-f002:**
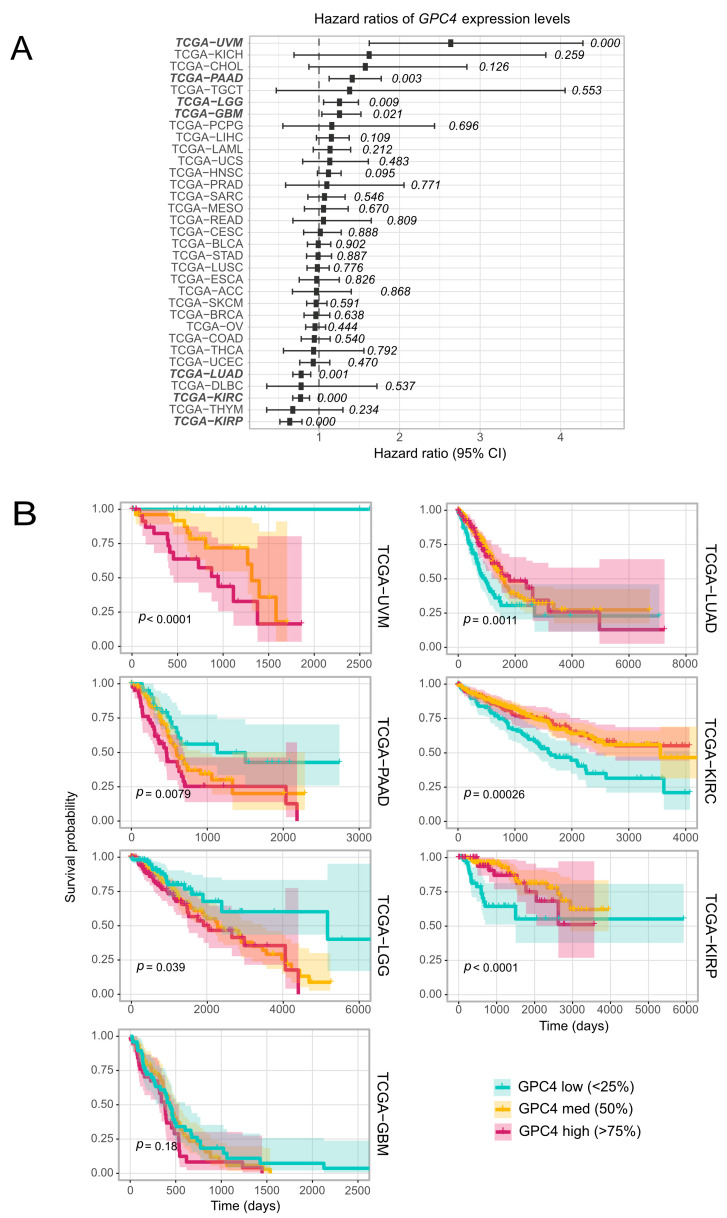
Association between *GPC4* expression and cancer prognosis. (**A**) Hazard ratio of *GPC4* expression levels on overall survival across TCGA cancer types. Cox proportional hazard p values are denoted next to each error bar. Error bars represent 95% confidence intervals. (**B**) Kaplan-Meier curves of the survival probability of *GPC4* expression strata in indicated TCGA projects in (**A**) where expression levels are associated with significant changes in cancer outcome. List of TCGA cancer abbreviations has been provided in Appendix A.

**Figure 3 ijms-25-03945-f003:**
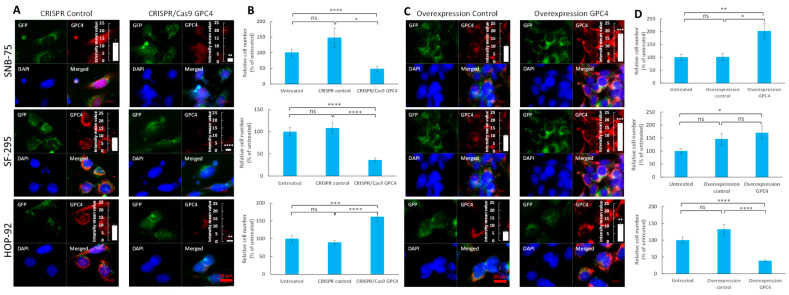
Suppression of *GPC4* expression attenuates proliferation of glioblastoma and augments proliferation of lung adenocarcinoma cells, whereas overexpression of *GPC4* augments proliferation of glioblastoma and attenuates proliferation of lung adenocarcinoma cells. (**A**,**B**) Depletion of *GPC4* expression by CRISPR/Cas 9 and (**C**,**D**) overexpression of *GPC4* by overexpression vector, in SNB-75, SF-295 and HOP-92 cells (as indicated in the images). (**A**) Depletion of endogenous *GPC4* expression by CRISPR/Cas9, as measured by immunofluorescence microscopy (60× magnifications). Cells were transfected with a control double nickase plasmid (not targeting any known gene; CRISPR Control) or a CRISPR/Cas9 double nickase plasmid construct targeting *GPC4* (CRISPR/Cas9 GPC4). (**C**) Overexpression of *GPC4* by overexpression vector, as measured by immunofluorescence microscopy (60× magnifications). Cells were transfected with a pCMV3-C-GFPSpark negative control vector not targeting any known gene (Overexpression Control) or a pCMV3-C-GFPSpark *GPC4* overexpression vector (Overexpression GPC4). Both CRISPR and overexpression constructs as well as control vectors encoded a GFP reporter to visualize successful transfection. After fixation with acetone the cells were stained with GPC4 antibody followed by Alexa Fluor 594-tagged goat anti-rabbit IgG. Expression of double nickase plasmids (GFP) and silencing or overexpression of *GPC4* (Alexa Fluor 594) was monitored by fluorescence microscopy. To visualize the cells, their nuclei were counterstained with DAPI (blue). The same exposure time was used in all experiments. Bar, 20 μm. Insets in (**A**,**C**): The intensity of GPC4 signal was determined in 4 identical low-magnification immunofluorescence images (20× magnification) and expressed in diagrams as mean intensity values ± SE. The level of immuno-reactive GPC4 was significantly suppressed in the CRISPR/Cas9 GPC4 cells and increased in GPC4-overexpressed cells as compared to controls (Student’s *t*-test, two-tailed unequal variances, *n* = 4, * *p* ≤ 0.05, ** *p* ≤ 0.01, *** *p* ≤ 0.001 and **** *p* ≤ 0.0001). (**B**,**D**) Effect of disruption of *GPC4* expression on proliferation of SNB-75, SF-295, and HOP-92 cells. SNB-75, SF-295, and HOP-92 cells were transfected with either (**B**) CRISPR control or CRISPR/Cas9 GPC4 vector or (**D**) overexpression control vector or *GPC4* overexpression vector as indicated in the images. After 3 days of proliferation, the cell densities were determined. Controls cells were left untreated cells containing only culture medium. The relative cell numbers were calculated as % of untreated cells. The graphs show results for double experiments (*n* = 6 in each experiment). Means ± SE are shown for each data point. The proliferation rate of SNB-75 and SF-295 cells was significantly decreased and the proliferation rate of HOP-92 cells was significantly increased in the CRISPR/Cas9 GPC4 transfected cells in comparison with the control cells (Student’s *t*-test, two-tailed unequal variances, *n* = 6). In contrast, the proliferation of SNB-75 cells exhibited a significant increase, while the proliferation of HOP-92 cells showed a significant decrease when transfected with the GPC4 overexpression vector in comparison with the control vector (Student’s *t*-test, two-tailed unequal variances, *n* = 6). *p* ≤ 0.05 was considered as statistically significant. *p* values are indicated as following: ns (not significant) *p* > 0.05, * *p* ≤ 0.05, ** *p* ≤ 0.01, *** *p* ≤ 0.001, and **** *p* ≤ 0.0001.

**Figure 4 ijms-25-03945-f004:**
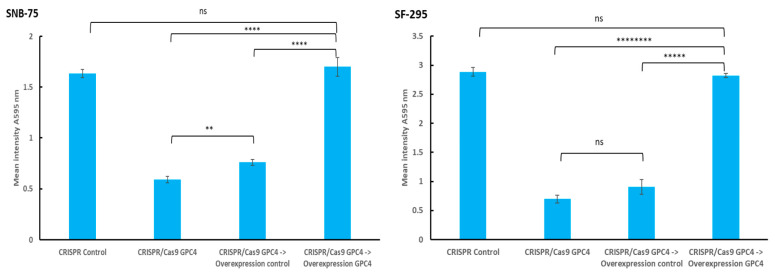
Overexpression of *GPC4* restores previously high proliferative rate following *GPC4* knockdown in gliobalstoma cells. Graphs show mean staining intensities measured at A595 nm for four different treatments (CRISPR Control, CRISPR/Cas9 GPC4, CRISPR/Cas9 GPC4 followed by overexpression control vector and CRISPR/Cas9 GPC4 followed by *GPC4* overexpression vector; *n* = 6 in each experiment). Means ± SE are shown for each experiment. The proliferative rate was significantly decreased for both SNB-75 and SF-295 cells treated with CRISPR/Cas9 GPC4 or CRISPR/Cas9 GPC4 followed by the overexpression control vector compared to cells treated with CRISPR/Cas9 GPC4 followed by the *GPC4* overexpression vector (Student’s *t*-test, two-tailed unequal variances, *n* = 6). Differences in staining intensities when comparing cells treated with CRISPR control or CRISPR/Cas9 GPC4 followed by *GPC4* overexpression vector were non-significant for both cell lines (Student’s *t*-test, two-tailed unequal variances, *n* = 6). *p* ≤ 0.05 was considered as statistically significant. *p* values are indicated as follows: ns (not significant) *p* > 0.05, ** *p* ≤ 0.01, **** *p* ≤ 0.0001, ***** *p* ≤ 0.00001 and ******** *p* ≤ 0.00000001.

**Figure 5 ijms-25-03945-f005:**
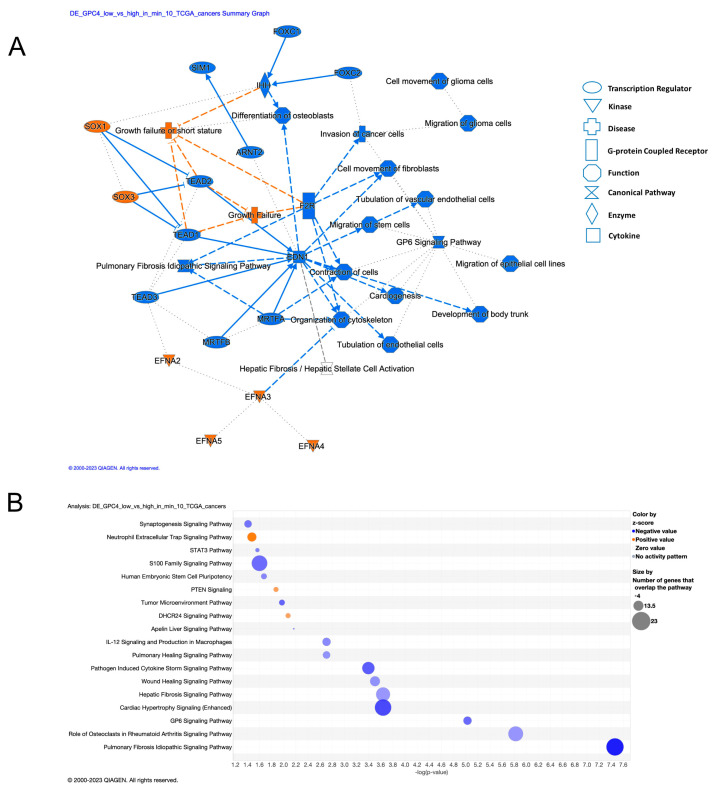
Pathway analyses of 352 genes that exhibited differential expression patterns in at least 10 cancer types between patients with high and low *GPC4* expression levels revealed the involvement of mechanisms associated with cancer cell proliferation, migration, and immunological aspects of cancer. (**A**) Network reconstruction of the predicted molecular relationships as inferred from gene expression changes between *GPC4*-high and *GPC4*-low cancer patients (orange: predicted activation; blue: predicted inhibition). (**B**) Dotplot presentation illustrating the top 10 canonical pathways enriched by enrichment ratio (*x*-axis). Color-coded predicted activity *z*-score where *z*-score > 0 indicates activation and *z*-score < 0 indicates inhibition. Pathways with negligible or no prediction (near-zero or no or prediction) are shown in gray. The size of the dots corresponds to the Benjamini-Hochberg adjusted *p* value (−log10).

**Figure 6 ijms-25-03945-f006:**
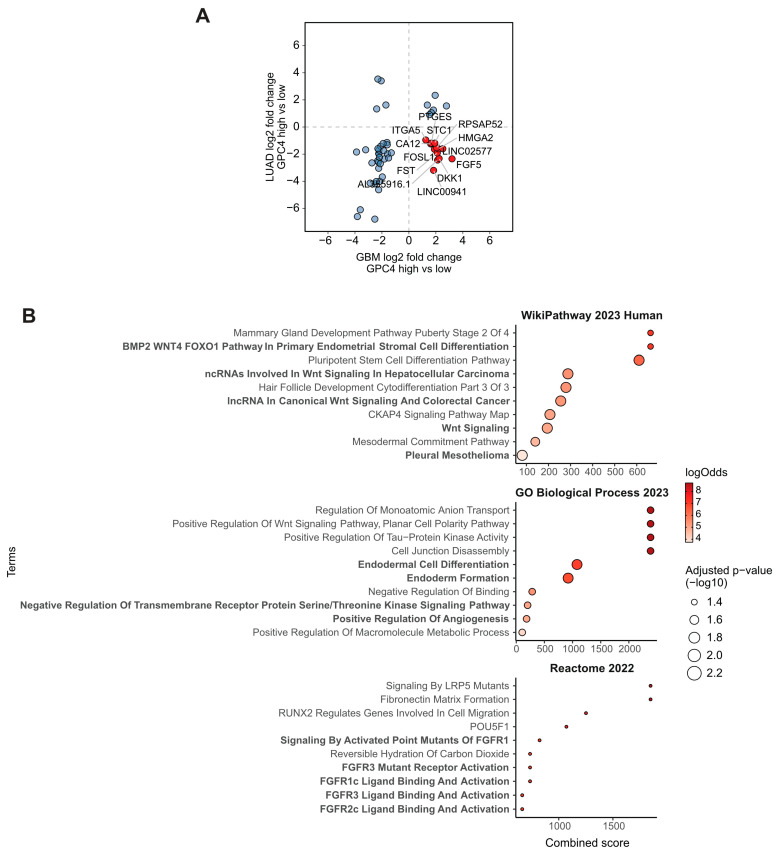
Discordant gene expression profiles explain the differential effects of *GPC4* upregulation between lung adenocarcinoma and glioblastoma. (**A**) Scatterplot depicting the log2 fold changes of genes found differentially expressed both in TCGA-LUAD (*y*-axis) and TCGA-GBM (*x*-axis) subjects between *GPC4*-high and *GPC4*-low cohorts. Genes that are upregulated in glioblastoma but downregulated in lung adenocarcinoma are highlighted in red. (**B**) Top 10 significant terms from pathway enrichment analysis results from 3 databases of the 12 differentially expressed genes with discordant profiles between TCGA-LUAD and TCGA-GBM. Combined enrichment score is plotted on *x*-axis. Enrichment log-odds over background is encoded as the bullet color while the bullet size encodes the enrichment adjusted *p* value (−log10).

**Table 1 ijms-25-03945-t001:** List of significant (adjusted *p* < 0.05) canonical pathways enrichment results.

Ingenuity Canonical Pathways	−log (*p*-Value)	*z*-Score	Ratio	Gene Names	*p*-Value	Adj. *p*-Value
S100 Family Signaling Pathway	2.62	−2.83683	0.0272	*CACNA1C*, *CACNA1H*, *EDNRA*, *EDNRB*, *FGFR1*, *FZD1*, *IGHG1*, *LGR5*, *MMP13*, *MMP7*, *NTRK2*, *PLA2G2A*, *PTH1R*, *RYR2*, *S100A2*, *S100A8*, *S100A9*, *S1PR3*, *SMAD9*, *TGFB2*, *WNT6*	0.00239	0.02515
Neutrophil Extracellular Trap Signaling Pathway	2.43	3.050851	0.0326	*COL10A1*, *COL11A1*, *COL12A1*, *COL1A1*, *COL21A1*, *COL25A1*, *COL26A1*, *COL2A1*, *COL6A3*, *COL8A1*, *COL8A2*, *IGHG1*, *PLA2G2A*	0.003715	0.03352
Myelination Signaling Pathway	2.24	−1.50756	0.0336	*ARHGAP6*, *BMP5*, *FGFR1*, *FZD1*, *HES5*, *ITGB3*, *NTRK2*, *PDGFRA*, *SMAD9*, *SOX8*, *WNT6*	0.005754	0.04962
Synaptogenesis Signaling Pathway	2.36	−2.7136	0.0349	*ADCY2*, *CADM1*, *CDH11*, *CDH6*, *CPLX2*, *EPHA3*, *EPHA7*, *NTRK2*, *RELN*, *THBS1*, *THBS4*	0.004365	0.03849
RAR Activation	3.49	−1.5	0.0373	*ADCY2*, *ADH1B*, *COL10A1*, *COL1A1*, *CRABP1*, *DKK2*, *FABP5*, *FGF10*, *MMP13*, *PDE1C*, *PDE3A*, *PDE5A*, *PRDM16*, *RBP4*, *RHOV*, *TGFB2*	0.000324	0.00502
Sperm Motility	2.51		0.0389	*CACNA1H*, *DDR2*, *EPHA3*, *FGFR1*, *NTRK2*, *PDE1C*, *PDGFRA*, *PLA2G2A*, *PRKG1*, *ROR1*	0.00309	0.03074
Cardiac Hypertrophy Signaling (Enhanced)	5.16	−3.44124	0.0406	*ADCY2*, *CACNA1C*, *EDNRA*, *EDNRB*, *FGF10*, *FGFR1*, *FZD1*, *GHR*, *IL17RD*, *ITGA8*, *ITGA9*, *ITGB3*, *MYOCD*, *PDE1C*, *PDE3A*, *PDE5A*, *PLN*, *PRKG1*, *RYR2*, *TGFB2*, *TNFRSF11B*, *WNT6*	0.000007	0.00022
RHOGDI Signaling	2.45	1.341641	0.0409	*ACTG2*, *ARHGAP6*, *CDH11*, *CDH6*, *ITGA8*, *ITGA9*, *ITGB3*, *MYH11*, *RHOV*	0.003548	0.03352
Activin Inhibin Signaling Pathway	2.49	−1.66667	0.0415	*CCN2*, *COL10A1*, *COL1A1*, *COL2A1*, *IGHG1*, *MMP7*, *SMAD9*, *TGFB2*, *TNFRSF11B*	0.003236	0.03138
Axonal Guidance Signaling	5.59		0.0431	*ADAM33*, *BMP5*, *CXCL12*, *DPYSL5*, *EPHA3*, *EPHA7*, *FZD1*, *ITGA8*, *ITGA9*, *ITGB3*, *MMP13*, *MMP7*, *NTN1*, *NTN4*, *NTRK2*, *SDC2*, *SEMA3D*, *SEMA3E*, *SLIT2*, *SLIT3*, *UNC5A*, *WNT6*	0.000003	0.0001
Gustation Pathway	2.7	−1	0.0446	*ABCC9*, *ADCY2*, *CACNA1C*, *CACNA1H*, *KCNJ8*, *KCNN2*, *LPL*, *PDE3A*, *SCN7A*	0.001995	0.02150
Human Embryonic Stem Cell Pluripotency	2.72	−2.33333	0.0448	*BMP5*, *FGFR1*, *FZD1*, *NR0B1*, *NTRK2*, *PDGFRA*, *SMAD9*, *TGFB2*, *WNT6*	0.001905	0.02112
Hepatic Fibrosis Signaling Pathway	5.15	−2	0.0449	*CACNA1C*, *CCN2*, *COL10A1*, *COL1A1*, *COL2A1*, *EDNRA*, *FGFR1*, *FZD1*, *ITGA8*, *ITGA9*, *ITGB3*, *MMP13*, *MYLK3*, *PDGFRA*, *RHOV*, *TF*, *TGFB2*, *TNFRSF11B*, *WNT6*	0.000007	0.00023
Pathogen-Induced Cytokine Storm Signaling Pathway	4.77	−3.15296	0.0458	*COL10A1*, *COL11A1*, *COL12A1*, *COL1A1*, *COL21A1*, *COL25A1*, *COL26A1*, *COL2A1*, *COL6A3*, *COL8A1*, *COL8A2*, *CXCL12*, *CXCL5*, *CXCL6*, *RYR2*, *TGFB2*, *TNFRSF11B*	0.000017	0.00041
Clathrin-mediated Endocytosis Signaling	3.2		0.0481	*ACTG2*, *ALB*, *AMPH*, *APOA2*, *APOC3*, *FGF10*, *ITGB3*, *RBP4*, *S100A8*, *TF*	0.000631	0.00844
Acute Phase Response Signaling	2.97	−1	0.0486	*ALB*, *APOA2*, *APOH*, *C4A/C4B*, *CRABP1*, *LBP*, *RBP4*, *TF*, *TNFRSF11B*	0.001072	0.01299
IL-12 Signaling and Production in Macrophages	3.97	−2.3094	0.0511	*ALB*, *APOA2*, *APOC3*, *COL10A1*, *COL1A1*, *COL2A1*, *DDR2*, *IGHG1*, *RBP4*, *S100A8*, *TGFB2*, *THBS1*	0.000107	0.00198
STAT3 Pathway	2.57	−2.44949	0.0519	*FGFR1*, *GHR*, *HGF*, *IL17RD*, *NTRK2*, *PDGFRA*, *TGFB2*	0.002692	0.02748
Tumor Microenvironment Pathway	3.07	−3	0.0503	*COL1A1*, *CXCL12*, *FGF10*, *HGF*, *ITGB3*, *MMP13*, *MMP7*, *TGFB2*, *TNC*	0.000851	0.01065
PTEN Signaling	2.93	2.236068	0.053	*FGFR1*, *GHR*, *ITGA8*, *ITGA9*, *ITGB3*, *NTRK2*, *PDGFRA*, *PREX2*	0.001175	0.01340
Semaphorin Neuronal Repulsive Signaling Pathway	2.95	0.377964	0.0533	*DPYSL3*, *DPYSL5*, *ITGA8*, *ITGA9*, *ITGB3*, *PRKG1*, *SEMA3E*, *VCAN*	0.001122	0.01319
Pulmonary Healing Signaling Pathway	3.99	−2.11058	0.0553	*CXCL12*, *FGF10*, *FGFR1*, *FZD1*, *MMP13*, *MMP7*, *SMAD9*, *TGFB2*, *THBS1*, *TNFRSF11B*, *WNT6*	0.000102	0.00198
Wound Healing Signaling Pathway	4.95	−2.13809	0.0556	*COL10A1*, *COL11A1*, *COL12A1*, *COL1A1*, *COL21A1*, *COL25A1*, *COL26A1*, *COL2A1*, *COL6A3*, *COL8A1*, *COL8A2*, *KRT16*, *TGFB2*, *TNFRSF11B*	0.000011	0.00031
Role of Osteoblasts, Osteoclasts, and Chondrocytes in Rheumatoid Arthritis	4.75		0.057	*BMP5*, *COL1A1*, *DKK2*, *FRZB*, *FZD1*, *ITGB3*, *MMP13*, *SFRP4*, *SFRP5*, *SMAD9*, *TNFRSF11B*, *WIF1*, *WNT6*	0.000018	0.00041
WNT/β-catenin Signaling	3.81	0.632456	0.0575	*DKK2*, *FRZB*, *FZD1*, *MMP7*, *SFRP4*, *SFRP5*, *SOX8*, *TGFB2*, *WIF1*, *WNT6*	0.000155	0.00261
DHCR24 Signaling Pathway	3.2	2.12132	0.0584	*ALB*, *APOA2*, *APOC3*, *APOH*, *C4A/C4B*, *RBP4*, *TF*, *VTN*	0.000631	0.00844
Cellular Effects of Sildenafil (Viagra)	3.63		0.06	*ACTG2*, *ADCY2*, *CACNA1C*, *KCNN2*, *MYH11*, *PDE1C*, *PDE3A*, *PDE5A*, *PRKG1*	0.000234	0.00379
Role of Osteoblasts in Rheumatoid Arthritis Signaling Pathway	5.8	−0.7746	0.0615	*COL1A1*, *CTSE*, *CXCL12*, *DKK2*, *FRZB*, *FZD1*, *MMP13*, *MMP7*, *SFRP4*, *SFRP5*, *SMAD9*, *TGFB2*, *TNFRSF11B*, *WIF1*, *WNT6*	0.000002	0.0001
Role of Osteoclasts in Rheumatoid Arthritis Signaling Pathway	7.94	−2.06474	0.0649	*ADAM33*, *COL10A1*, *COL11A1*, *COL12A1*, *COL1A1*, *COL21A1*, *COL25A1*, *COL26A1*, *COL2A1*, *COL6A3*, *COL8A1*, *COL8A2*, *FRZB*, *ITGB3*, *MMP13*, *MMP7*, *RHOV*, *SFRP4*, *SFRP5*, *TNFRSF11B*	0.00000001	0.0000015
White Adipose Tissue Browning Pathway	3.9	−1.66667	0.0652	*ADCY2*, *CACNA1C*, *CACNA1H*, *FGFR1*, *FNDC5*, *PRDM16*, *PRKG1*, *RUNX1T1*, *VGF*	0.000125893	0.00222
Caveolar-mediated Endocytosis Signaling	2.43		0.0667	*ACTG2*, *ALB*, *ITGA8*, *ITGA9*, *ITGB3*	0.003715352	0.03352
Osteoarthritis Pathway	6.72	−0.33333	0.0678	*COL10A1*, *COL2A1*, *DDR2*, *FGFR1*, *FRZB*, *FZD1*, *ITGA8*, *ITGA9*, *ITGB3*, *MMP13*, *PTH1R*, *RBP4*, *S100A8*, *S100A9*, *S1PR3*, *SMAD9*	0.0000002	0.000009
Microautophagy Signaling Pathway	4.86		0.0688	*COL10A1*, *COL11A1*, *COL12A1*, *COL1A1*, *COL21A1*, *COL25A1*, *COL26A1*, *COL2A1*, *COL6A3*, *COL8A1*, *COL8A2*	0.000014	0.00036
Pulmonary Fibrosis Idiopathic Signaling Pathway	9.75	−4.26401	0.0706	*ACTG2*, *CCN2*, *COL10A1*, *COL11A1*, *COL12A1*, *COL1A1*, *COL21A1*, *COL25A1*, *COL26A1*, *COL2A1*, *COL6A3*, *COL8A1*, *COL8A2*, *CXCL12*, *EDNRA*, *FGFR1*, *FZD1*, *MMP13*, *MMP7*, *PDGFRA*, *TGFB2*, *THBS1*, *WNT6*	0.0000000002	0.00000003
Maturity Onset Diabetes of Young (MODY) Signaling	3.12		0.0759	*ABCC9*, *APOA2*, *APOC3*, *APOH*, *CACNA1C*, *KCNJ8*	0.000759	0.00981
FXR/RXR Activation	5.01		0.0794	*ALB*, *APOA2*, *APOC3*, *APOH*, *C4A/C4B*, *FABP6*, *LPL*, *RBP4*, *TF*, *VTN*	0.00001	0.00029
Netrin Signaling	3.33	−0.8165	0.0833	*CACNA1C*, *CACNA1H*, *NTN1*, *PRKG1*, *RYR2*, *UNC5A*	0.000467735	0.00687
Dilated Cardiomyopathy Signaling Pathway	6.78	−1.26491	0.0867	*ABCC9*, *ACTG2*, *ADCY2*, *CACNA1C*, *CACNA1H*, *CNN1*, *DES*, *MYH11*, *PDE3A*, *PLN*, *RYR2*, *SGCD*, *TNNC1*	0.0000002	0.000009
GP6 Signaling Pathway	6.72	−2.88675	0.0945	*COL10A1*, *COL11A1*, *COL12A1*, *COL1A1*, *COL21A1*, *COL25A1*, *COL26A1*, *COL2A1*, *COL6A3*, *COL8A1*, *COL8A2*, *ITGB3*	0.0000002	0.000009
LXR/RXR Activation	6.87	1.507557	0.0976	*ALB*, *APOA2*, *APOC3*, *APOH*, *C4A/C4B*, *LBP*, *LPL*, *RBP4*, *S100A8*, *TF*, *TNFRSF11B*, *VTN*	0.0000001	0.000009
Atherosclerosis Signaling	7.4		0.0977	*ALB*, *ALOX12B*, *APOA2*, *APOC3*, *COL10A1*, *COL1A1*, *COL2A1*, *CXCL12*, *LPL*, *MMP13*, *PLA2G2A*, *RBP4*, *S100A8*	0.00000004	0.000004
Hepatic Fibrosis/Hepatic Stellate Cell Activation	13.5		0.113	*CCN2*, *COL10A1*, *COL11A1*, *COL12A1*, *COL1A1*, *COL21A1*, *COL25A1*, *COL26A1*, *COL2A1*, *COL6A3*, *COL8A1*, *COL8A2*, *EDNRA*, *EDNRB*, *FGFR1*, *HGF*, *LBP*, *MMP13*, *MYH11*, *PDGFRA*, *TGFB2*, *TNFRSF11B*	0.00000000000003	0.00000000001
Intrinsic Prothrombin Activation Pathway	4.66	−0.8165	0.143	*COL10A1*, *COL1A1*, *COL2A1*, *F12*, *F5*, *KLK10*	0.00002	0.00047
Apelin Liver Signaling Pathway	3.32	−2	0.148	*COL10A1*, *COL1A1*, *COL2A1*, *EDN3*	0.000478	0.00688
Role of IL-17A in Psoriasis	4.5	0	0.286	*CXCL5*, *CXCL6*, *S100A8*, *S100A9*	0.00003	0.00065

## Data Availability

The datasets associated with the findings in this work are available from the corresponding author upon reasonable request.

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
