# Peer review of "Dichotomous Effects of Glypican-4 on Cancer Progression and Its Crosstalk with Oncogenes"

_ijms, 2024, doi:10.3390/ijms25073945_

Round 1

Reviewer 1 Report

Comments and Suggestions for Authors

In this work, the authors performed functional genomics assays in glioblastoma and non-small cell lung cancer in vitro models to understand how changes in GPC4 expression affect, revealing divergent cancer type-dependent effects on proliferation. They then couple these results with survival analysis of public cancer data allowing the authors to link clinical outcomes with GPC4 expression-associated changes in cancer growth. Finally, systematic transcriptomic analysis of cancer patients uncovered divergent gene expression profiles between glioblastoma and lung adenocarcinoma subjects exhibiting pleiotropic effects on regulation of pathways linked to oncogenic signaling.

In summary, the manuscript's findings unravel the divergent effect of GPC4 on cancer progression and clinical outcome, particularly in glioblastoma and non-small cell lung cancer and illuminates the pleiotropic effect of GPC4 on proto-oncogene singling pathways that govern mitogenic behavior. The manuscript was very well-written, very comprehensive, and the experiments were very carefully designed and performed. All the backend data was also provided in detail. After all, I think this paper can be published with the current form. And here are just some minor comments:

1. In Supplementary document, the CRISPR/Cas9 GPC4 anti-GPC4 antibody and anti-tubulin antibody slot blot overexpression assays were hard to read. The bands are unclear, could the authors make these gel images clearer? 

2. Similarly, the Supplementary Figure 1B, for the secondary antibody image, there is no difference shown between the overexpression control and the overexpression GPC4. The image is very blurry. Could the authors make this better? Thanks!

Author Response

Responses to comments from the Reviewers:

Reviewer 1:

We thank the Reviewer for the supportive comments. A point-to-point response to the Reviewer’s comments is provided below.

  1. In Supplementary document, the CRISPR/Cas9 GPC4 anti-GPC4 antibody and anti-tubulin antibody slot blot overexpression assays were hard to read. The bands are unclear, could the authors make these gel images clearer? 

Response: We appreciate this comment. In response, we have upgraded the slot blot figure to a higher resolution version. Additionally, the supplementary figure has been expanded and intensity measurements has been incorporated, providing additional confirmation of GPC4 CRISPR/Cas9 knockdowns and overexpressions (please see supplementary Figure 1, specifically panels B and D)."

  1. Similarly, the Supplementary Figure 1B, for the secondary antibody image, there is no difference shown between the overexpression control and the overexpression GPC4. The image is very blurry. Could the authors make this better? Thanks!

Response: Secondary antibody control showed no detectable signal, indicating absence of background staining. To clarify this issue, we have removed the secondary antibody image and included the information in the figure legend instead stating: ’No signal was detected when the primary antibody was omitted.'

Reviewer 2 Report

Comments and Suggestions for Authors

This manuscript describes the role of GPC4 in regulating tumor cell proliferation and highlights the potential of treating it as a biomarker for tumor prognosis. The novel aspects of the study include triangulating TCGA gene expression and survival datasets to identify certain genes that may play important roles in controlling prognosis / survival. However, beyond strengths, there are also caveats that authors need to address in order to advance the study.

1. It is interesting to see the pleiotropic effect of GPC4, but the tumor-specific function of GPC4 might dampen the potential of GPC4 as a biomarker to predict prognosis. for example, in PAAD, GPC4 is downregulated, associated with a high survival rate, while in lung and kidney, upregulation of GPC4 leads to high survival. These 2 opposite trends necessitate detailed explanation of why this might be the case, despite biology of cancer might vary across different tumor types. I would like to understand fundamentally why GPC4 could play opposite roles in different cancer types.

2. Figure 3B SNB-75, CRISPR control increased cell count from 100 to 150 by looking at the absolute number, while in A549, control also decreased cell count from 100 to 75, despite these changes were insignificant. Authors claimed CRISPR control had no effect on the cell proliferation, but I recommend authors repeat the experiment and make sure the control indeed had no effect on proliferation. Besides, on NIH website, A549 and SF-295 cell lines proliferate with similar rate. Therefore, the statement that KD of GPC4 in A549 did not show increased proliferation was because cell already proliferate at max speed was not convincing, unless we also didn't see increased proliferation in SF-295 when overexpressing GPC4.

2. In order to mitigate some non-specific effects, suggest authors can also perform some rescue experiment. For example, they can overexpress GPC4 on top of knock down or knock down GPC4 after overexpression to see if cell proliferation can be brought back to the control level (Figure 3B, D). 

3. In figure 4, 5, authors used multiple databases to identify some potential downstream targets of GPC4 that mediate the control of cell proliferation. However, would like to see some follow-up functional study. will be helpful if authors can validate the findings with experiment (e.g., knock down certain downstream targets and measure cell proliferation again). Without functional study, we all know signals in GO analysis or other pathway identification analysis might be dirty with off-target noise.

Comments on the Quality of English Language

This manuscript was well written in English with only minor revision needed to improve the grammar and quality.

Author Response

Responses to comments from the Reviewers:

Reviewer 2:

We thank the Reviewer for the supportive comments. 

A point-to-point response to the Reviewer’s comments is provided below.

  1. It is interesting to see the pleiotropic effect of GPC4, but the tumor-specific function of GPC4 might dampen the potential of GPC4 as a biomarker to predict prognosis. for example, in PAAD, GPC4 is downregulated, associated with a high survival rate, while in lung and kidney, upregulation of GPC4 leads to high survival. These 2 opposite trends necessitate detailed explanation of why this might be the case, despite biology of cancer might vary across different tumor types. I would like to understand fundamentally why GPC4 could play opposite roles in different cancer types.

Response: We appreciate the reviewer's supportive comments and acknowledge the valid point that the potential of GPC4 as a biomarker for predicting prognosis cannot be universally generalized across all cancers, given the documented dual and opposing effects on cancer progression. In response, we have made modifications in the abstract and discussion sections, mentioning the biomarker potential of GPC4 specifically in certain cancer types (please see abstract, last sentence; Results, page 3, last paragraph; and Discussion, page 13, last paragraph).

While the fundamental understanding of why GPC4 may exhibit opposite roles in different cancer types is indeed an intriguing question, we believe that a conclusive answer is beyond the scope of our current investigation and may emerge in the future.

  1. Figure 3B SNB-75, CRISPR control increased cell count from 100 to 150 by looking at the absolute number, while in A549, control also decreased cell count from 100 to 75, despite these changes were insignificant. Authors claimed CRISPR control had no effect on the cell proliferation, but I recommend authors repeat the experiment and make sure the control indeed had no effect on proliferation. Besides, on NIH website, A549 and SF-295 cell lines proliferate with similar rate. Therefore, the statement that KD of GPC4 in A549 did not show increased proliferation was because cell already proliferate at max speed was not convincing, unless we also didn't see increased proliferation in SF-295 when overexpressing GPC4.

Response: We have addressed this issue by conducting experiments detailed in supplementary Figure 2. In these experiments, A549 cells were transfected with increasing concentration of CRISPR/Cas9 and CRISPR control vectors which involve either doubling (supplementary Figure 2A) or quadrupling their concentrations (supplementary Figure 2B). Increasing the concentration of CRISPR/Cas9 GPC4 led to a notable and significant increase in the proliferation rate of A549 cells compared to untreated cells (Supplementary Figure 2; P values: 0.001 and 0.002 respectively). However, CRISPR/Cas9 GPC4 had not a significant impact on the proliferation rate of A549 cells compared to CRISPR controls (Supplementary Figure 2; P values: 0.284 and 0.813 respectively).

In response to this comment and following our new results, this section has undergone substantial revision where we have removed the statement that “knock-down of GPC4 in A549 did not show increased proliferation was because cell already proliferate at max speed” and incorporated in the new results instead (Please see page 7, last paragraph and page 8 first paragraph).

  1. In order to mitigate some non-specific effects, suggest authors can also perform some rescue experiment. For example, they can overexpress GPC4 on top of knock down or knock down GPC4 after overexpression to see if cell proliferation can be brought back to the control level (Figure 3B, D). 

Response: We thank the reviewer for suggesting this experiment, which we agree would be valuable together with an appropriate control. However, control experiment with withdrawal of CRISPR/Cas9 alone could also lead to increased cell proliferation. Consequently, interpreting the extent to which the effect on cell proliferation is a response to overexpression on the top of CRISPR/Cas9 or CRISPR/Cas9 withdrawal would be challenging, not providing clear answers about the non-specific effects.

  1. In figure 4, 5, authors used multiple databases to identify some potential downstream targets of GPC4 that mediate the control of cell proliferation. However, would like to see some follow-up functional study. will be helpful if authors can validate the findings with experiment (e.g., knock down certain downstream targets and measure cell proliferation again). Without functional study, we all know signals in GO analysis or other pathway identification analysis might be dirty with off-target noise.

Response: We agree that our interpretations may carry a degree of speculation. Respectfully, we differ in opinion regarding the efficacy of the suggested experiments involving the knockdown of specific downstream targets and subsequent measurement of cell proliferation. We contend that such experiments, particularly targeting protooncogenes such as FGF5, TGF-beta, or ITGA-5, would likely result in a decrease in cell proliferation. However, it is crucial to note that such outcomes would primarily underscore the importance of these protooncogenes in cancer, rather than provide substantive information about their connection with GPC4. Finally, we would like to express that our interpretations are grounded in reason and not merely fanciful.

Round 2

Reviewer 2 Report

Comments and Suggestions for Authors

Appreciate authors' prompt response and revision. Here are my detailed response to each point:

1. agreed it is convoluted to answer the reason for the pleitropic effect of certain gene, so the current statement works for me

2. Im still not convinced with the data demonstrated in supplementary figure 2. Authors claimed that GPC4 has no effect on A549 cell proliferation. but if we look at the absolute value of cell count in Figure 3B and Supplementary Figure 2. Figure 3B, both CRISPR control and GPC4 knockdown decreased cell number while in Supp Fig 2 both CRISPR control and GPC4 knockdown increased cell number. These 2 results are controversial. In Figure 3D, overexpression did suppress cell proliferation in A549. I think authors need to think thoroughly about the mechanism of GPC4 in A549, otherwise would recommend removing the findings in A549.

3. If authors claim withdrawal of CRISPR/Cas9 it self caused disturbance to cell proliferation, then I would argue this type regulation is not specific to GPC4 itself, but rather it is under the control of CRISPR/Cas9 vector introduction. This will void the whole point of the paper. Unless authors can show evidence that overexpression of GPC4 in GPC4 CRISPR knocked down cell lines can mitigate the knockdown effect, any effects authors claim here specific to GPC4 are not convincing

4. I think functional studies of any computational analysis that tries to identify connections between signaling pathways are critical because there are caveats to each database in terms of nonspecific effects. As long as the sample size is large enough, any 2 random things in the world can be proved statistically interconnected. But whether the connection is functionally meaningful that is the value of molecular cancer study. I understand there might be technical barriers authors worry before conducting these experiments, but that is also somewhere authors can add value and novelty to the study. 

Comments on the Quality of English Language

Language is not the barrier, but I believe the results authors added in the revision are somehow contradictory and less convincing to me. 

Author Response

Dear Editor,

We would like to express our appreciation to the Referee for the prompt response.

We, hereby, resubmit a revised version of our manuscript together with detailed response to the referee's comments, incorporating adjustments and additional experiments as requested. However, we wish to address Comment 4, which requests experimental validation of the computational analysis within the 10-day response period allotted by the journal. Regrettably, meeting this request within the given timeframe is not feasible.

During the initial revision process, time constraints was also a challenge, preventing us from adequately planning and executing the requested experiments. Despite this, we immediately initiated the rescue experiments proposed by the reviewer in Comment 3. Unfortunately, these experiments were not concluded before the previous response period elapsed. However, we are now pleased to present the results in this revision.

With these factors in mind, we are resubmitting our revisions and hope that the referee will consider the substantial improvements made. A detailed response to the Referee's comments is provided below, and all changes have been highlighted in the manuscript.

Best regards,

Katrin Mani, Corresponding author

Responses to comments from the Reviewers:

Comments 1. agreed it is convoluted to answer the reason for the pleiotropic effect of certain gene, so the current statement works for me.

Response: We appreciate the referee’s positive response.

  1. Im still not convinced with the data demonstrated in supplementary figure 2. Authors claimed that GPC4 has no effect on A549 cell proliferation. but if we look at the absolute value of cell count in Figure 3B and Supplementary Figure 2. Figure 3B, both CRISPR control and GPC4 knockdown decreased cell number while in Supp Fig 2 both CRISPR control and GPC4 knockdown increased cell number. These 2 results are controversial. In Figure 3D, overexpression did suppress cell proliferation in A549. I think authors need to think thoroughly about the mechanism of GPC4 in A549, otherwise would recommend removing the findings in A549.

Response: We appreciate your concern regarding the observed discrepancies in the A549 cell proliferation data. We acknowledge the conflicting results, which indeed indicate inconsistency in the behavior of A549 cells. Despite our efforts to elucidate the underlying mechanism of GPC4 in A549 cells, we have been unable to reconcile these conflicting results. In light of your recommendation we have removed the A549 findings from the manuscript.

  1. If authors claim withdrawal of CRISPR/Cas9 itself caused disturbance to cell proliferation, then I would argue this type regulation is not specific to GPC4 itself, but rather it is under the control of CRISPR/Cas9 vector introduction. This will void the whole point of the paper. Unless authors can show evidence that overexpression of GPC4 in GPC4 CRISPR knocked down cell lines can mitigate the knockdown effect, any effects authors claim here specific to GPC4 are not convincing.

Response:  Thank you for your valuable feedback and your patience as the suggested experiment has enhanced the quality of our manuscript.

Regarding the assumption that withdrawal of CRISPR/Cas9 itself would affect cell proliferation, our reasoning was based on the transient nature of the CRISPR/Cas9 GPC4. However, we understood the need for experimental validation. Therefore, we immediately initiated the requested experiment in SNB-75 and SF-295 cells. Despite our efforts, the results were not concluded before the response period for previous revision elapsed. We are now pleased to report that overexpression of GPC4 in CRISPR/Cas9 GPC4 knocked down cells indeed mitigates the knockdown effect. Importantly, as a control, CRISPR/Cas9 GPC4 cells were transfected with an overexpression negative control vector to ensure the specificity of our observations. These findings are presented in a dedicated Figure 4 and detailed in the "Results" section (2.5, page 8), highlighted in yellow. Please see also yellow in “Abstract” and “Discussion”.

  1. I think functional studies of any computational analysis that tries to identify connections between signaling pathways are critical because there are caveats to each database in terms of nonspecific effects. As long as the sample size is large enough, any 2 random things in the world can be proved statistically interconnected. But whether the connection is functionally meaningful that is the value of molecular cancer study. I understand there might be technical barriers authors worry before conducting these experiments, but that is also somewhere authors can add value and novelty to the study. 

Response: Thank you for your insightful comment regarding the importance of functional studies to validate computational analyses. We acknowledge the value of experimental validation in adding robustness to our study. Indeed, we are actively planning for experimental validation of the computational analysis to ensure the functional relevance of the identified connections between signaling pathways. However, we must also acknowledge the technical and logistical challenges associated with conducting these experiments within the 10-day response period allotted by the journal. Given the complexity and scope of the experimental validation required, completing these studies in such a short timeframe would not be feasible. While, we understand and respect the importance of upholding the standards, we would appreciate your consideration. Thank you once again for your valuable feedbacks and understanding.